# Preparation of Zirconium Phosphate Nanomaterials and Their Applications as Inorganic Supports for the Oxygen Evolution Reaction

**DOI:** 10.3390/nano10050822

**Published:** 2020-04-26

**Authors:** Mario V. Ramos-Garcés, Jorge L. Colón

**Affiliations:** 1Department of Chemistry, University of Puerto Rico, 17 Ave. Universidad STE 1701, San Juan, PR 00925-2537, USA; mario.ramos1@upr.edu; 2NSF PREM Center for Interfacial Electrochemistry of Energy Materials, University of Puerto Rico, San Juan, PR 00925, USA

**Keywords:** zirconium phosphate, oxygen evolution reaction, electrocatalysis, inorganic nanomaterials

## Abstract

Zirconium phosphate (ZrP) nanomaterials have been studied extensively ever since the preparation of the first crystalline form was reported in 1964. ZrP and its derivatives, because of their versatility, have found applications in several fields. Herein, we provide an overview of some advancements made in the preparation of ZrP nanomaterials, including exfoliation and morphology control of the nanoparticles. We also provide an overview of the advancements made with ZrP as an inorganic support for the electrocatalysis of the oxygen evolution reaction (OER). Emphasis is made on how the preparation of the ZrP electrocatalysts affects the activity of the OER.

## 1. Introduction

The first report of a crystalline form of zirconium phosphate dates back to 1964 [1]. The chemical composition of this material was determined to be Zr(HPO_4_)_2_·H_2_O. Later, the first crystal structure of this crystalline ZrP material was reported in 1969 and then refined in 1977 [2,3]. Its structure consists of a layered arrangement (Figure 1). Zr(IV) ions align in a near perfect plane bridged by orthophosphate groups, which are 5.3 Å apart from each other, above and below the Zr ions plane. Each Zr ion is coordinated by six oxygen atoms from six different phosphate groups, forming an octahedral coordination with the metal center. Three of the oxygen atoms from each phosphate group are coordinated to three different Zr ions. The fourth oxygen is bonded to a hydrogen atom and points above and below each ZrP layer. The stacking of these layers, which are 6.6 Å thick and 7.6 Å apart from each other, creates a zeolitic cavity with a diameter of 2.61 Å that is occupied by a water molecule [4,5]. This description corresponds to that of a phase that has been named α-ZrP, the most extensively studied phase of ZrP. Other phases have been achieved by varying the synthesis protocol and some of these phases will be explained later in this review.

ZrP acidity and porosity are desired traits for different applications, especially in heterogeneous catalysis [6,7]. Not only that, but these parameters are tunable, expanding the possible applications of ZrP [8]. Beyond that, the major attractiveness of ZrP comes from its ability to perform ion-exchange with its acidic phosphate groups. The composite materials that result after ion-exchange differ greatly in their properties when compared to α-ZrP, allowing them to be implemented for several different applications. Ion-exchange in ZrP occurs at the Brönsted acid groups (P-OH) which are also present at the surface of the nanoparticles, opening another pathway for the modification of this material: surface modification [9,10,11]. These composite materials have been used for several applications including photocatalysis [12,13], drug delivery [14,15,16,17], amperometric biosensors [18,19,20], catalysis [7,21,22,23,24,25,26], flame retardancy [27,28,29], and others. We encourage the reader to also see other recent reviews on the topic of the synthesis of ZrP and its applications [7,8,30,31]. 

## 2. Synthesis and Preparation of ZrP Nanomaterials

### 2.1. α-ZrP

Clearfield and Stynes were able to synthesize the first crystalline ZrP (α-ZrP) by refluxing amorphous gels in phosphoric acid [1]. Later, a hydrothermal method and a synthesis with HF were reported [32]. The method used for the synthesis of α-ZrP has a direct impact on the size and shape of the resulting nanoparticles. This is another advantage as we can adjust these parameters depending on the application while still retaining the main characteristics of α-ZrP. More specifically, the concentrations of reactants, temperature during synthesis, pressure, and use of complexing agents have been found to have an effect on the aspect ratios of the nanoparticles [33]. 

Sun et al. reported the synthesis of α-ZrP nanoparticles through three different methods and characterized the materials [34]. The first method consisted of mixing ZrOCl_2_·8H_2_O with phosphoric acid (H_3_PO_4_) at varying concentrations. Namely, 3, 6, 9, and 12 M H_3_PO_4_ at 100 °C and 24 h. The second method consisted of mixing ZrOCl_2_·8H_2_O with H_3_PO_4_ (at the same concentrations as in approach 1) in a Teflon^®^ pressure vessel and heating each reaction at 200 °C for 24 h. The final approach consisted of mixing ZrOCl_2_·8H_2_O with 3.0 M H_3_PO_4_ in a Teflon^®^ flask. HF was added so that the molar ratios of F/Zr were 1, 2, 3, and 4.

The authors of this study observed that, for method 1, the crystallinity and aspect ratio increase as the H_3_PO_4_ concentration increases. The resulting nanoplatelets had sizes ranging from ~60 nm to ~200 nm. The nanoplatelets that resulted from method 2 had an enhanced crystallinity when compared to those from method 1. Moreover, the aspect ratio increases and the size of the nanoplatelets now range from ~400 to ~1200 nm as the concentration of H_3_PO_4_ increases. Finally, the α-ZrP platelets obtained from method 3 have an even higher crystallinity than the nanoplatelets from method 2. An increase in the aspect ratio is also observed and the sizes range from ~2000 to over ~4000 nm. The scanning electron microscopy (SEM) images of these samples are shown in Figure 2. Recently, Contreras-Ramirez et al. made a detailed study on how the synthesis method and reaction conditions affect the structural order and crystallinity of α-ZrP. In their study, the authors used synchrotron X-ray atomic pair distribution function (PDF) analysis to observe the changes in the structural order caused from the different methods and conditions [35]. 

The reflux, hydrothermal, and HF method for the synthesis of α-ZrP have been the most widely used methods throughout the decades. However, other methods have been reported in the literature such as precipitation with oxalic acid and by liquid-phase deposition [36,37,38]. Pica et al. reported the synthesis via an alcohol intercalation/deintercalation method [39]. α-ZrP was prepared by dissolving zirconium propionate in different anhydrous alcohols (ethanol, propanol, and butanol) and adding H_3_PO_4_ at different H_3_PO_4_/Zr molar ratios (2, 4, and 6). This synthesis procedure results in transparent gels. Characterization of these gels by X-ray powder diffraction (XRPD) shows only one well defined peak in all diffractograms corresponding to (002) planes of ZrP. This corresponds to the interlayer distance and it was found to vary depending on the alcohol used during synthesis. More specifically, the interlayer distance was found to be 14.4 Å when ethanol was used, 16.1 Å for propanol, and 18.6 Å for butanol. This highlights that the alcohols are contained within the layers of ZrP. The TEM images of these samples reveal that the resulting nanoparticles are hexagonal shaped and have regular planar sizes of ~40 nm, independent of which alcohol was used. When the gels are dried at 60 °C until complete solvent evaporation, a white powder is obtained. The XRPD diffractograms for all samples shows a shift in the first peak to d = 7.56 Å, indicating the formation of α-ZrP. The authors found that the crystallinity of the resulting α-ZrP nanoparticles was dependent on the H_3_PO_4_/Zr molar ratio and increases with increasing ratio. The transmission electron microscopy (TEM) images for the samples prepared with an H_3_PO_4_/Zr ratio of 6 show that the nanoparticles kept a more or less hexagonal morphology and the distribution of planar sizes increased to a ~30–200 nm range.

Highly crystalline α-ZrP was reported with a new method by using a minimal solvent synthesis procedure [40]. By adding H_3_PO_4_ to powder ZrOCl_2_·8H_2_O with a H_3_PO_4_/Zr molar ratio of two, α-ZrP was successfully obtained. Moreover, if the H_3_PO_4_/Zr molar ratio is increased to three, α-ZrP with enhanced crystallinity is obtained. The SEM images of α-ZrP prepared with a H_3_PO_4_/Zr molar ratio of three show that the cross-sectional dimensions of the nanoplatelets are in the range of ~100–500 nm with a thickness of ~40–66 nm. This method introduces a greener and milder option for the synthesis of highly crystalline α-ZrP as no excess of H_3_PO_4_ is required and/or the use of hazardous HF.

### 2.2. ZrP Exfoliation

The process of separating the layers of a bulk layered material is known as exfoliation. This process can be achieved by a variety of methods and has been extensively studied for several layered materials [41]. The two-dimensional materials (2D) of nanosheets or few layered nanosheets that result after exfoliation have been shown to have several advantages over their bulk systems [42]. Enhancement in conductivity, surface area, mechanical flexibility, optical transparency, and other properties make these exfoliated materials suitable for application in different areas, such as electrocatalysis, electronics, energy storage, and others [42]. 

ZrP has been successfully exfoliated and its nanosheets used for different applications [6,21,43,44,45,46,47,48,49,50]. The most widely used strategy for ZrP exfoliation consists of the intercalation of small amine cations that can easily displace the protons from the phosphate groups in an acid-base reaction and enter the interlayer space [51]. The mechanism of this process is said to be the formation of an amine double layer in the interlayer space, leading to exfoliation due to cation–cation repulsions [44]. Kaschak et al. studied the exfoliation of ZrP with tetra-*n*-butylammonium hydroxide (TBA^+^OH^−^) [52]. The authors estimated the TBA^+^ diffusion rate within the ZrP galleries from the time required to achieve a constant expansion rate and the distance travelled in this time. The intercalation rates determined suggests a first-order process that depends on the opening of the interlayer at the edges of the nanoplatelets. As the exfoliation of ZrP is known to produce hydrolysis products, Kaschak et al. studied this process by atomic force microscopy (AFM) and TEM. Results show that the hydrolysis reaction occurs from the edges inward and the percent hydrolysis increases as a function of time. The hydrolysis process after 1 h was monitored by varying the temperature of the reaction. It was found that the hydrolysis percent varied drastically with temperature. For both semi-crystalline and micro-crystalline ZrP, the rate of hydrolysis during exfoliation is essentially zero at 0 °C. However, at higher temperatures, the percent hydrolysis is noticeable and increases with increasing temperature.

ZrP has also been successfully exfoliated via a melt-compounding method [53]. Here, ZrP is first intercalated with diglycolamine (DGA). After intercalation, the product was dried and ground into fine powders and subsequently mixed with maleic anhydride grafted polyolefin elastomer (POE-g-MA). The mixtures were then exfoliated with melt compounding.

Another method that has been reported for the exfoliation of ZrP is the assisted exfoliation with ionic liquids [54]. Herein, α-ZrP is first intercalated with small molecules such as DGA. Then, the DGA intercalated ZrP is mixed with the ionic liquid 1-methyl-3-*n*-octylimidazolium bromide [OMIm]Br and ultrasonicated for 30–60 min. After product collection and characterization, the authors noticed the successful exfoliation of ZrP. The initial intercalation step is crucial as it helps in expanding the interlayer spacing and weakens the H-bonding between the layers. DGA is thought to attach to the ZrP through its amine end while the rest of the molecule is a small polar chain that can attract OMIm cations. Hence, it is believed that OMIm enters the interlayer through a cation-lone pair attraction, forming highly charged surfaces, which leads to the complete exfoliation of α-ZrP and stabilization of the nanosheets in ionic liquids.

Recently, an ZrP exfoliation procedure using Tris-(hydroxymethyl)-aminomethane (Tris) was reported [48]. For this, α-ZrP is added to a Tris buffer solution and the mixture is ultrasonicated for 20 min. This method introduces the substitution of commonly used organic strong bases to a more environmentally friendly, less basic, and non-toxic Tris exfoliating agent.

After exfoliation, some exfoliating agents can be displaced with another cationic species if the latter is put in contact with a suspension of the exfoliated ZrP nanoparticles. To obtain the protonated ZrP nanosheets, producing the Brönsted acid groups, a follow-up method with an acid can be performed [21]. The exfoliation of ZrP has also been accomplished by using alkanol amines and other wet methods [55,56]. 

### 2.3. ZrP with Different Morphologies

By adding fluoride anions (F^−^) in the synthesis of α-ZrP using the minimal solvent synthesis procedure, crystal growth along the a-axis is favored [40]. The result of this is the formation of α-ZrP particles with rod-like morphology. The addition of different F^−^ sources, such as NaF and NH_4_F, and keeping the synthesis temperature and duration at 100 °C and 24 h, respectively, results in highly crystalline rod-like particles along with platelet-like structures. The observed platelets have sizes around 1 μm and the rods are 4–10 μm in length. Reaction time studies show that increasing the duration of the reaction favors the formation of the rods over the platelet structures (Figure 3). It is believed that the formation of zirconium–fluoro complexes on the (001) crystal planes due to selective absorption of the fluoride ions is the cause of the favored a-axis crystal growth.

Pica et al. reported that cube-like ZrP nanoparticles can be obtained with the alcohol intercalation/deintercalation procedure [57]. When the gels are dried at 120 °C instead of 60 °C, a different crystal structure is obtained. The phase resulting from this synthesis method that produces cube-like ZrP was found to be similar to a previously reported phase of ZrP called τ-ZrP, with a few discrepancies. For this reason, this new phase was labelled as τ’-ZrP. This phase was described to have Zr octahedrally coordinated by six HPO_4_ tetrahedra, forming eight-membered rings. These eight-membered rings contain two P-OH groups pointing to the same side of the Zr plane. The 3D structure of τ’-ZrP consists of the packing of these planes perpendicularly to each other (Figure 4). These authors reported that the formation of τ’-ZrP from α-ZrP prepared by this method is possible by heating α-ZrP at 120 °C. It was also observed that the transformation of τ’-ZrP to α-ZrP occurs at room temperature. However, when both products are washed with the alcohol used during synthesis, neither transformation occurs. It is believed that the presence of free H_3_PO_4_ catalyzes these transformations. Hence, washing the products removes the extra H_3_PO_4_ present and stabilizes the phases, preventing the reversibility between τ’-ZrP and α-ZrP. ZrP derivatives have also been successfully prepared with cube morphologies [58]. 

ZrP has also been successfully synthetized with sphere morphologies. Zhang et al. reported the synthesis of microspherical ZrP by a colloid mill procedure [59]. For this, α-ZrP is first prepared by dissolving ZrOCl_2_·8H_2_O in HCl and added along with a H_3_PO_4_/HCl acid solution to a colloid mill at 3000 rpm for 2 min. The resulting solution is then collected with 15 M H_3_PO_4_ and refluxed at 363 K for 72 h. To prepare the microspheres, the precipitate of the above solution is redispersed in H_2_O to form a precursor slurry. This slurry is then fed into an atomizer and an aerosol dispersion is generated within the tubular reactor of the spray-drying apparatus. After the shaping process with set parameters, the collected α-ZrP microspheres showed diameters in the range of 5–45 μm. The authors observed that calcination of the microspheres does not change the morphology.

The synthesis of smaller microspheres was reported by Tarafdar et al. [60]. These microspheres are mesoporous in nature and have diameters ranging from 1–3 μm. To prepare these mesoporous spheres, ZrOCl_2_·8H_2_O is mixed with (NH_4_)_2_CO_3_ and (NH_4_)_2_HPO_4_ in H_2_O. Tetradecyltrimethylammonium bromide (TTABr) is added as a surfactant and the mixture is heated at 80 °C for 3 d, followed by aging in an autoclave at 90 °C for 2 d, and at 120 °C for 1 d. The obtained product is then calcined at 540 °C for 6 h.

The synthesis of spherical ZrP has also been reported using a microwave-assisted hydrothermal method [61]. More specifically, ZrOCl_2_·8H_2_O is mixed with H_3_PO_4_ and placed in a microwave oven for 30 min at 120 °C. The obtained gel is then washed and dried at 80 °C for 24 h. Characterization with XRPD confirms that the product is α-ZrP. SEM images elucidate that the spheres are composed of an agglomeration of smaller α-ZrP nanoparticles.

ZrP has also been prepared with a flower-like morphology by a solid-state synthesis [62]. For this, ZrOCl_2_·8H_2_O is mixed Na_3_PO_4_ and ground together for 50 min. Then, the mixture is autoclaved at 80 °C for 96 h. XRPD confirms that α-ZrP was prepared and the SEM images indicate that these flower-like particles have dimensions of 5–7 μm. These flowers are composed of smaller ZrP nanosheets of ~100 nm in thickness. It was found that extending the reaction time beyond 96 h destroys the flower-like morphology and that it is only obtained at 80 °C. The phosphate precursor also plays an important role, as it was observed that using an alternate source did not produce the flower-like structure.

ZrP hexagonal prisms (Figure 5) were prepared by dissolving ZrOCl_2_·8H_2_O in formamide, mixing the solution with H_3_PO_4_ and heating the mixture in a hydrothermal reactor at 150 °C for 24 h [63]. This is followed by aging at predetermined temperatures (60, 80, 100, and 150 °C) for 1 week. The prisms grow due to the selective adsorption of ammonium cations (from the solvent) onto the layer surfaces that are parallel to the precipitated (001) crystal plane. This allows the growth of layers on top of each other. Further characterization shows that these ZrP prisms are NH_4_ZrH(PO_4_)_2_, which has an interlayer distance of 11.18 Å. Treating these prisms with acid converts them into α-ZrP. However, the prismatic structure is destroyed during this conversion. It is worth noting that ZrP and some derivatives have also been reported with different morphologies other than the platelet-like structures [64,65]. Table 1 summarizes the various methods of ZrP nanoparticle preparation reported herein.

### 2.4. Intercalation of Guest Species into ZrP

Whittingham defined intercalation (in chemistry) as “the reversible insertion of guest species into a lamellar host structure with maintenance of the structure features of the host” [66]. The zeolitic cavity with a diameter of 2.61 Å in α-ZrP impedes the direct intercalation of species with larger dimensions, hence the intercalation of these species is not significant and/or they are exchanged at very slow rates [5,67,68,69]. To circumvent this problem, pre-intercalation methods such as the intercalation of sodium cations or small alkyl amines into α-ZrP (producing expanded ZrP phases) are commonly performed as the first step for the intercalation of the intended guest species [69,70]. However, these pre-intercalation methods typically result in the co-intercalation of various species, hindering the analysis of experimental results.

To address this, Martí and Colón reported in 2003 a new method for the direct intercalation of large metal complexes that does not require a pre-intercalation step [13]. This method consists of using a highly hydrated phase of zirconium phosphate, namely, Zr(HPO_4_)_2_·6H_2_O (θ-ZrP) [13]. θ-ZrP can be prepared by using a reflux method that results in a material with the same type of layers as α-ZrP, but with an interlayer distance of 10.4 Å, instead of 7.6 Å [71]. The increased interlayer distance is due to the six water molecules per formula unit in θ-ZrP, in contrast with α-ZrP that only has one [72]. When θ-ZrP is allowed to dry, it converts back to α-ZrP. This can be confirmed with XRPD as the first diffraction peak at 2θ = 8.6° (d_002_ = 10.4 Å) for θ-ZrP, which corresponds to the interlayer distance, shifts towards 11.6° (d_002_ = 7.6 Å) when this material dehydrates. Hence, determining if an intercalation was successful is trivial, as the XRPD analysis of a dry intercalation product should yield a first diffraction peak with a distance greater than 7.6 Å [14].

In their study, Martí and Colón intercalated the luminescent metal complex tris(2,2’-bipyridyl ruthenium(II) ([Ru(bpy)_3_]^2+^) by direct ion exchange into ZrP, using θ-ZrP (Figure 6) [13]. The intercalated product shows an increase in the interlayer distance to 15.2 Å, and [Ru(bpy)_3_]^2+^ retains its structural integrity. This method of intercalation via direct ion exchange in ZrP is by an ion exchange reaction with the protons in the Brönsted acid groups (P-OH). As the intercalation proceeds, the intercalant enters the interlayer space though the edges and diffuses into the center of the solid [73]. The direct intercalation of several guest species into θ-ZrP has been successfully used to produce composite materials for different applications [18,20,74,75,76,77,78,79].

## 3. ZrP Nanomaterials as Support for OER Active Species

The OER is the oxidative half reaction of water electrolysis, consisting of the four-electron oxidation of water to produce oxygen and protons. In alkaline media, the reaction mechanism of the OER is described with four oxidation steps (Equations (1)–(4), where AS stands for active site) [80]. The first step is the one electron oxidation of a hydroxide ion, followed by a proton-coupled electron transfer, giving an oxygen atom adsorbed onto the AS. The third step is a nucleophilic attack coupled to a one electron oxidation, producing an hydroperoxide species adsorbed onto the AS. The fourth and final step is a second proton-coupled electron transfer step and subsequent product removal from the AS. It is an important half-cell reaction in many energy-related schemes such as CO_2_ reduction, metal-air batteries, and photoelectrochemical (PEC) water splitting [81]. However, the OER suffers from considerable overpotential losses due to its sluggish kinetics, hindering the large-scale implementation of sustainable clean energy production [82]. Benchmarking studies of electrocatalytic systems for the OER have suggested that their overpotentials have plateaued, limiting the efficiency of future PEC and other technological devices [83,84,85]. In these studies, all the catalysts in alkaline solution require overpotentials of 0.30–0.50 V to drive the reaction at a current density of 10 mA/cm^2^. This may suggest common mechanistic limitations and new strategies are needed in order to produce catalysts that are more efficient and more stable for this reaction.
AS + OH^−^ → e^−^ + AS-OH(1)
AS-OH → e^−^ + H^+^ + AS-O(2)
AS-O + OH^−^ → e^−^ + AS-OOH(3)
AS-OOH → e^−^ + AS + H^+^ + O_2_(4)

Catalysis can be improved in two ways: increasing the number of actives site in a system or increasing the intrinsic activity of active sites (Figure 7) [86]. This catalytic improvement can be accomplished by a variety of experimental methods such as intercalation/confinement, supporting active species onto supports, nanostructuring, and many others. ZrP high thermal and chemical stability, as well as its electrochemical inertness, makes it a suitable material to be studied as an OER support. The electrochemical inertness is of outmost importance for a material to be considered a potential support and ZrP is known to not present oxidative behavior at the potential range of OER studies [87]. The layered structure of ZrP makes it possible to study the OER in a confined environment when actives species are intercalated in it. The presence of the Brönsted acid groups P-OH at the surface of the nanoparticles, makes it possible to study the reaction at ZrP-modified surfaces. Moreover, ZrP versatility in controllable synthesis to produce several different morphologies and nanoparticles with different aspect ratios makes it suitable for nanostructuring studies. Nanostructuring ZrP supporting structures could improve the loading of active species, expose more active sites to the electrolyte and/or increase the intrinsic activity of each active site by changes in synergy between the active species and support. In this section, the recent advancements made with ZrP as an inorganic support for transition metals for the electrocatalysis of the OER will be overviewed. 

### 3.1. Intercalated and Surface Adsorbed Transition Metals ZrP Electrocatalysts

Sanchez et al. performed the first study of ZrP as a support for earth-abundant transition metal cations for the OER [87]. On this study, the transition metal cations Fe^2+^, Fe^3+^, Co^2+^, and Ni^2+^ were either intercalated into or adsorbed onto α-ZrP. To intercalate the transition metals, the authors utilized the direct ion exchange method developed by Martí and Colón. For this, a suspension of θ-ZrP is mixed with a solution of the metal salt precursor and left stirring for several days at room temperature so that ion-exchange reaches equilibrium (Figure 8A). Ion-exchange in ZrP occurs at the Brönsted acid groups which are also present at the surface of the nanoparticles. Hence, there is no way of preventing metal cations adsorbing on the surface. To obtain more insight into the nature of the activity of the samples, a second metal-modified ZrP system in which the metals are only adsorbed onto the surface of the nanoparticles was also prepared. To prepare the surface-adsorbed counterparts, α-ZrP is used as the ZrP precursor instead of θ-ZrP (Figure 8B). The metal cations are large enough that intercalation into α-ZrP was not observed. Characterization through several methods such as XRPD, Fourier transform infrared spectroscopy (FT-IR), and thermogravimetric analysis (TGA) confirm the successful preparation of both metal-modified ZrP catalytic systems.

The electrochemical activity towards the OER of these systems was assessed with cyclic volatammetry (CV) experiments using a rotating disk electrode (RDE) in 0.1 M KOH electrolyte. The OER acitvity was determined from the overpotential necessary to achieve 10 mA/cm^2^, as this parameter has been suggested to be used to benchmark OER catalyst activity [84]. The overpotential at a current density of 10 mA/cm^2^ is the potential difference between the potential necessary to achieve that current density and the thermodynamic potential of water oxidation (1.23 V vs RHE). All ZrP cataytic systems were active for the OER, requiring between 0.5–0.7 V of overpotential to reach 10 mA/cm^2^. However, the authors observed that, in general, the OER activities for the metal-adsorbed ZrP catalysts are greater than or equal to those of their metal-intercalated counterparts at the same synthesis M:ZrP molar ratios, as seen by their lower overpotentials (Figure 9). This result highlights an important aspect of these systems: the OER occurs preferentially on the outer surfaces of the nanoparticles [87,88]. This is deduced from the fact that XPS measurements corroborate that the intercalated catalysts have a higher metal loading than those in which the metals are only adsorbed. Hence, the intercalated species are not electrochemically accessible to perform the OER which the authors hypothetized that it may be due to differences in mass transport and/or electronic conductivity in the two systems.

### 3.2. Exfoliation of ZrP for Improved OER

Following the previously obtained results in which the OER occurs preferentially on the surface of layered metal-modified ZrP nanoparticles, Ramos-Garcés et al. prepared metal-modified exfoliated ZrP electrocatalysts [47]. The authors exfoliated ZrP by using TBA^+^OH^−^ as the exfoliating agent and afterwards, did an acid wash with HCl to obtain ZrP nanosheets. These nanosheets were subsequently modified with Co^2+^ and Ni^2+^ cations at an excess 10:1 M:ZrP molar ratio with the aim of obtaining a maximum metal loading on the systems. They also prepared the previously reported Co^2+^ and Ni^2+^ surface-adsorbed ZrP electrocatalysts at the same excess molar ratio to compare their activity to the novel exfoliated materials.

The activity of these exfoliated nanomaterials was detemined through overpotential, Tafel slope, mass activity, and turnover frequency (TOF) basis. Results show that the exfoliated nanomaterials required less overpotential to achieve the same current density when compared to the surface-adsorbed layered nanoparticles (Figure 10). The intrinsic activity of each catalytic site was probed with mass activity and TOF analyses using the metal loading determined experimentally by inductively coupled plasma-mass spectrometry (ICP-MS). These results show that the lower overpotential of the exfoliated samples is due to a larger number of active sites. ICP-MS results show that ZrP nanosheets are substantially better at adsorbing Co^2+^ and Ni^2+^ cations, leading to higher loadings than non-exfoliated ZrP. However, these exfoliated catalysts maintain comparable intrinsic activity to metal-modified layered catalysts, resulting in higher geometric normalized activities due to the significant greater number of actives sites.

### 3.3. Nanostructuring ZrP to Support Transition Metals Species

Recently, Ramos-Garcés et al. reported the dependence of OER activty on the morphology of the metal-modified ZrP support [89]. These authors prepared ZrP materials with different morphologies: rod-like ZrP using the minimal solvent synthesis method [40], cube-like ZrP using the alcohol intercalation/deintercalation method [57], mesoporous spheres using the Tarafdar et al. method [60], and α-ZrP hexagonal platelets obtained from the reflux method. The ZrP supports with different morphologies were prepared for their use as supports for Co^2+^ and Ni^2+^ cations. Figure 11 shows the XRPD patterns of these ZrP structures [87]. The ZrP support structures were treated with a large excess (molar ratio) of the metal salt solutions, expecting to obtain similar loadings. However, different loadings were obtained, as evidenced by ICP-MS. Therefore, the number of active species on the surface of the metal-modified ZrP supports is different for the different morphologies. Co^2+^ cations exchanged with surface ZrP protons to a higher level than the Ni^2+^ cations, regardless of ZrP morphology. The maximum loading levels followed the trend α-ZrP > rod-like ZrP > cube-like ZrP > ZrP spheres, indicating that the different particle sizes and crystallinities affect the ion exchange behavior of the ZrP materials, as previously observed [35,90]. 

The Co-modified ZrP catalysts show an expected trend in activity, as the activity decreased with a decrease in Co-loading from support to support. The activity decrease was observed in overpotential, Tafel slope, and mass analysis basis. However, this trend is reversed for the Ni-based catalysts. That is, as the Ni content increased, the activity from support to support decreased. This observation was explained by the difference in the surface-area-to-volume-ratio (SA:V) of the different ZrP supports and the in-plane conductivity of the catalysts. The SA:V was calculated from the average dimensions of the ZrP structures that were obtained through TEM/STEM. The calculated SA:V for the different ZrP structures were: 0.07 (α-ZrP), 0.16 (ZrP rods), 0.034 (ZrP cubes), 0.0039 (ZrP spheres).

α-ZrP and rod-like ZrP, the ZrP morphologies with higher SA:V, achieved higher loadings of Co^2+^ and obtained a higher coverage compared to the other ZrP morphologies. The higher coverage of cobalt species improved the conductivity of the materials and the electrocatalysis performance. In contrast, the conductivity of the Ni-modified ZrP catalysts decreased with increased Ni^2+^ loading. When the SA:V is taken into consideration, it can be deduced that the higher coverage of Ni species in the ZrP rods and α-ZrP leads to abundant resistive nickel species. This results in inactive areas on the surface of the supporting structures due to conductivity issues that can hinder the transport of electrons for the composite catalysts during the OER. Interestingly, the Ni-modified ZrP spheres show a much higher activity compared to the other ZrP structures. This was explained by the much lower coverage of Ni species on the micron size particles, and issues with conductivity might be minimized.

The geometric normalized activities of these metal-modified ZrP materials are considered moderate. However, an analysis of intrinsic activity, such as the mass activity, shows the potential of the ZrP structures as supports for OER actives species. Table 2 shows the mass activities for the different Metal/ZrP catalysts at an overpotential of 350 mV, as well as for the IrO_x_ catalyst in alkaline electrolyte. These data show that the choice of the ZrP support tunes the mass activity for the different catalysts. However, it is interesting to note that a low loading Ni/ZrP catalyst showcases a mass activity higher than benchmark IrO_x_. This highlights the importance of the metal coverage and conductivity for these types of ZrP catalysts, making these parameters targets for improving their performance.

## 4. Conclusions

This review highlights some advancements made in the preparation of ZrP nanomaterials. This includes the synthesis of α-ZrP and how the synthesis method may affect its aspect ratio. We also overviewed exfoliation techniques and intercalation chemistry of ZrP. Synthesis strategies that have been explored to produce ZrP materials with different morphologies were also explored.

This work concludes with an overview of an emerging application of ZrP: supports for metal species for the electrocatalysis of the OER. Advancements to date have shown that conductivity plays an important role in the activity of these catalysts. Moreover, physical factors such as species coverage also need to be considered when trying to improve these systems. Studies with mixed-metal systems are underway. We envision that new strategies to improve OER catalysis with ZrP will show promising results.

## Figures and Tables

**Figure 1 nanomaterials-10-00822-f001:**
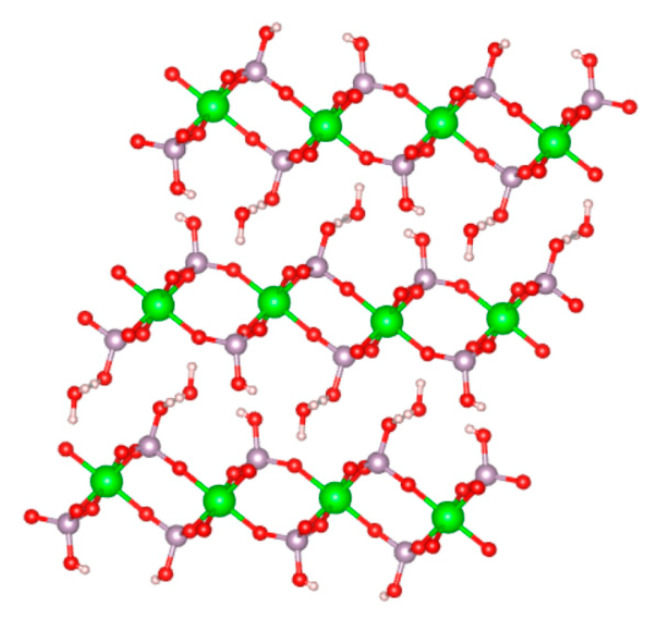
Ball and stick model of the structure of α-ZrP.

**Figure 2 nanomaterials-10-00822-f002:**
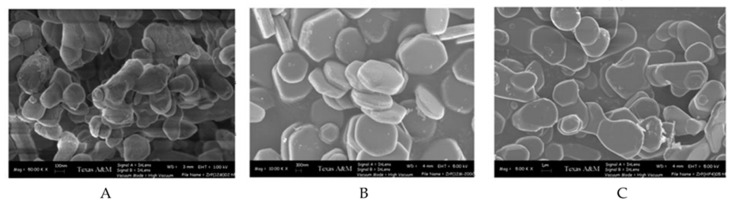
SEM images of α-ZrP prepared by (**A**) reflux method (12 M H_3_PO_4_) (**B**) hydrothermal method (9 M H_3_PO_4_) (**C**) HF method (F/Zr = 4). [34]—Adapted by permission of The Royal Society of Chemistry on behalf of the Centre National de la Recherche Scientifique, 2007.

**Figure 3 nanomaterials-10-00822-f003:**
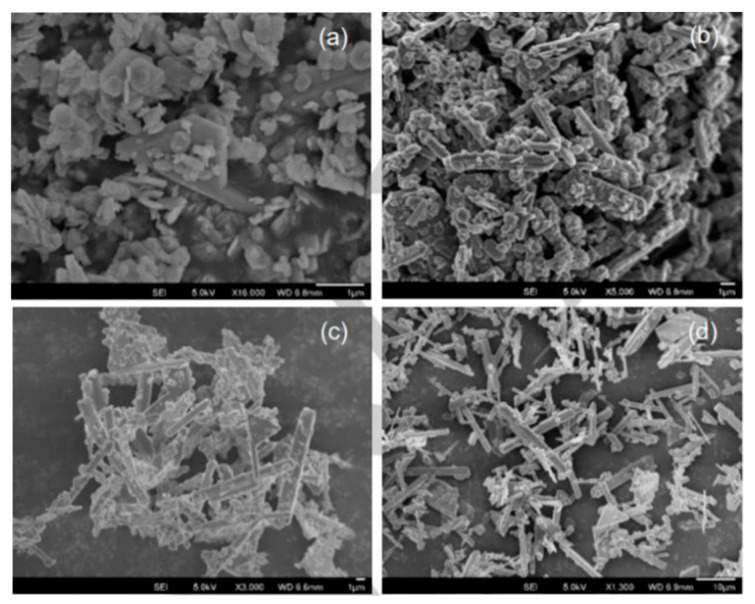
SEM images of α-ZrP prepared by the minimal solvent synthesis method with F^−^ ions at different reaction times; (**a**) 1 h (**b**) 4 h (**c**) 8 h and (**d**) 96 h. Reproduced from [40], with permission from John Wiley and Sons, 2017.

**Figure 4 nanomaterials-10-00822-f004:**
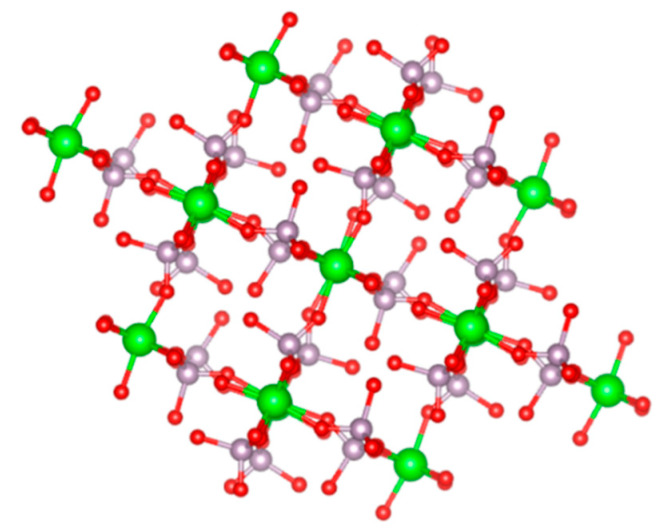
Ball and stick model of the structure of τ’-ZrP [57].

**Figure 5 nanomaterials-10-00822-f005:**
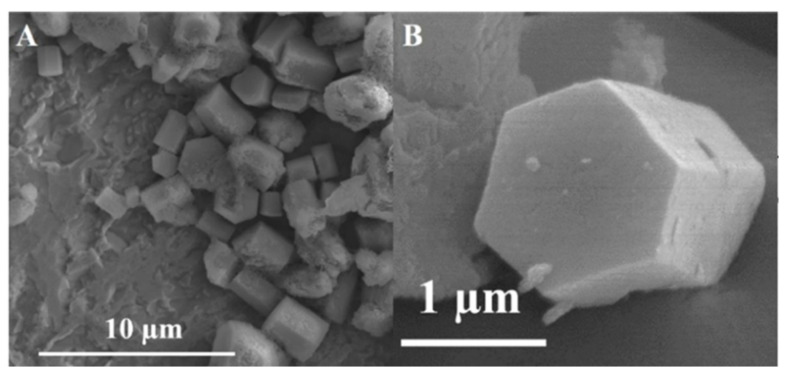
SEM images (**A**,**B**) of the prepared ZrP prisms. Reproduced from [63], with permission from the American Chemical Society, 2020.

**Figure 6 nanomaterials-10-00822-f006:**
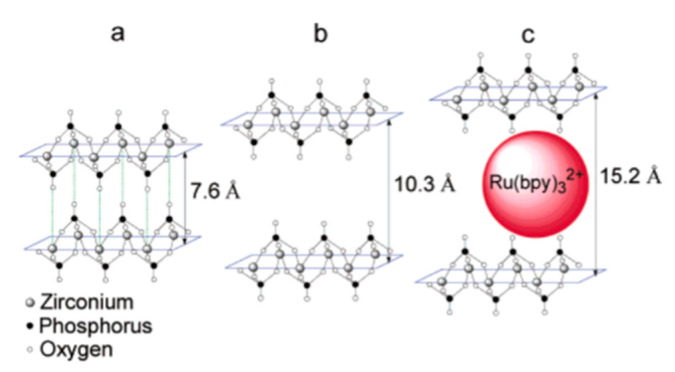
Schematic representation of (**a**) α-ZrP (**b**) θ-ZrP and (**c**) [Ru(bpy)_3_]^2+^-exchanged ZrP. Reproduced from [13], with permission from the American Chemical Society, 2003.

**Figure 7 nanomaterials-10-00822-f007:**
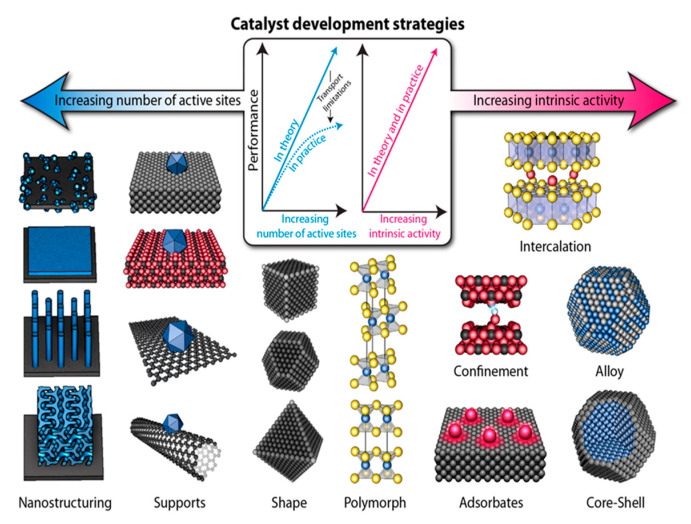
Schematic representation of various catalyst development strategies, which aim to increase the number of active sites and/or increase the intrinsic activity of each site. Reproduced from [86], with permission from the American Association for the Advancement of Science, 2017.

**Figure 8 nanomaterials-10-00822-f008:**
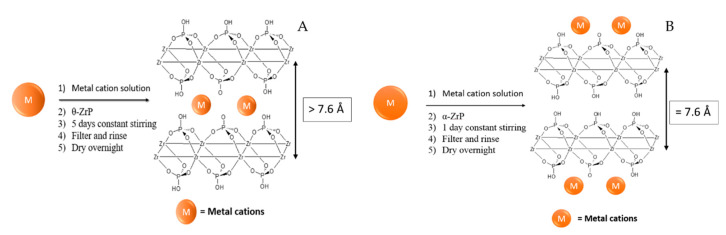
Schematic representation of the preparation procedure of (**A**) metal-intercalated and (**B**) surface adsorbed ZrP catalysts.

**Figure 9 nanomaterials-10-00822-f009:**
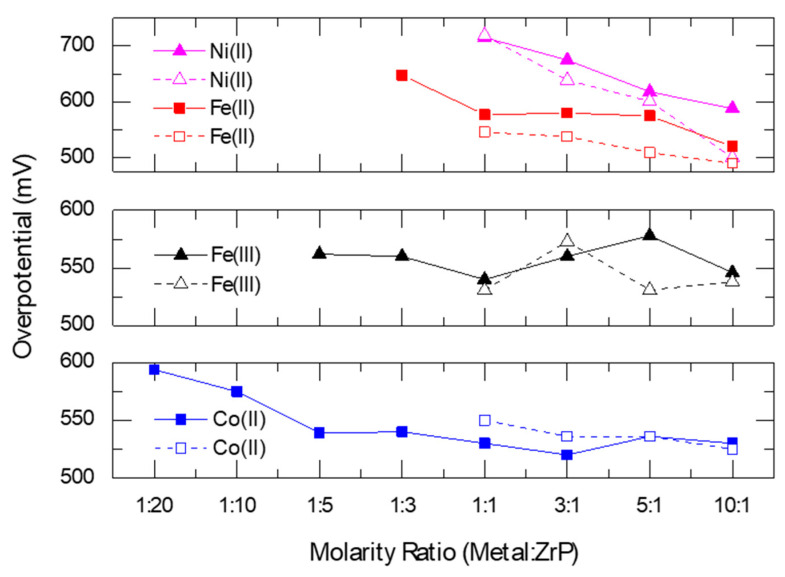
Overpotential as a function of M:ZrP synthesis ratio for the different metal-modified ZrP catalysts. Dashed lines correspond to surface-adsorbed catalysts while the solid line correspond to metal-intercalated catalysts. Taken from reference [87].

**Figure 10 nanomaterials-10-00822-f010:**
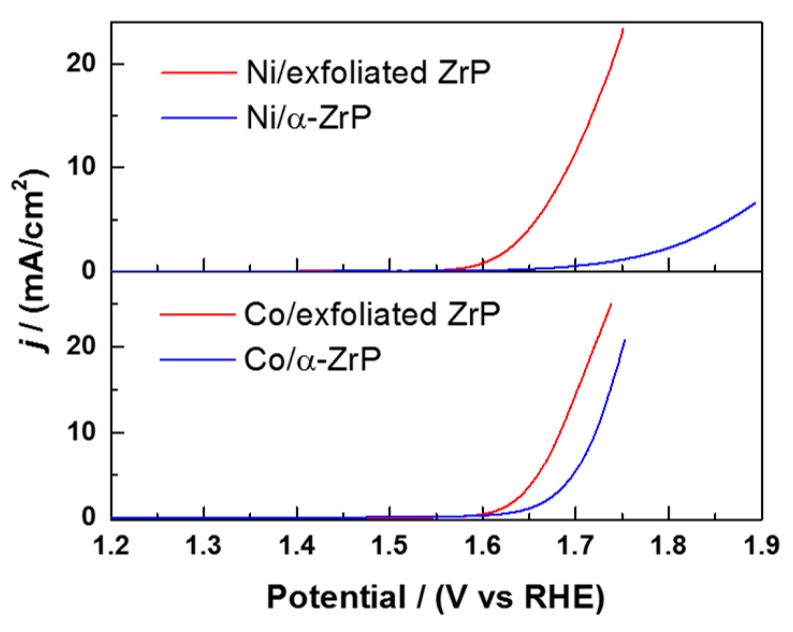
iR-compensated linear sweep voltammograms of Ni-modified and Co-modified ZrP catalysts in 0.1 M KOH. Reproduced from [47], with permission from the American Chemical Society, 2019.

**Figure 11 nanomaterials-10-00822-f011:**
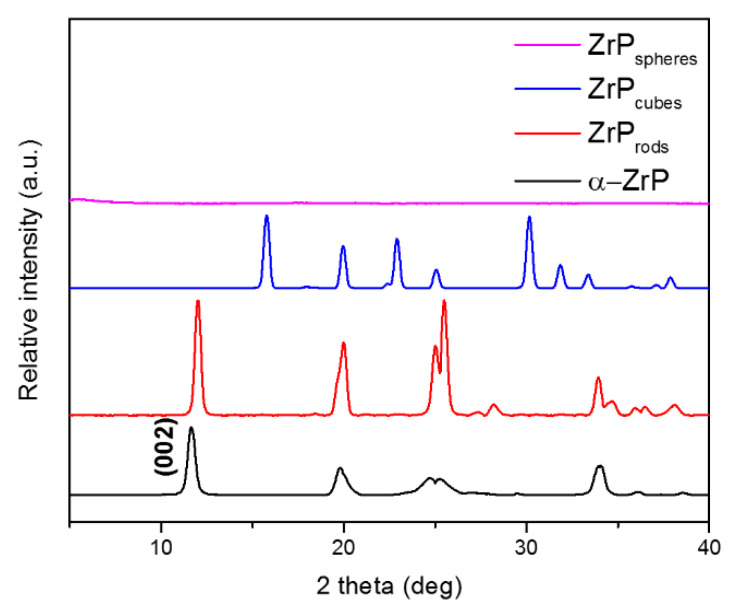
XRPD patterns of α-ZrP, rod-like ZrP, cube-like ZrP, and spherical ZrP. [89]—Reproduced by permission from The Royal Society of Chemistry.

**Table 1 nanomaterials-10-00822-t001:** Summarizing table on the different preparation methods of ZrP nanoparticles.

Method	Morphology	Size (nm)	Reference
Reflux	Hexagonal platelets	60–200	[34]
HF	Hexagonal platelets	2000–4000	[34]
Hydrothermal	Hexagonal platelets	400–1200	[34]
Oxalic acid precipitation	Hexagonal platelets	2000–3000	[37]
Liquid-phase deposition	Hexagonal platelets	2000–3000	[38]
Alcohol intercalation/deintercalation (60 °C)	Hexagonal platelets	30–200	[39]
Minimal solvent	Hexagonal platelets	100–500	[40]
Minimal solvent (with F^−^)	Rod-like	4000–10,000	[40]
Alcohol intercalation/deintercalation (120 °C)	Cube-like	100–500	[57]
Colloid mill procedure	Spheres	5000–445,000	[59]
Tarafdar et al.	Spheres	1000–3000	[60]
Microwave-assisted hydrothermal	Spheres	1000–3000	[61]
Mu et al.	Flower-like	5000–7000	[62]
Yu et al.	Hexagonal prisms	1000–3000	[63]

**Table 2 nanomaterials-10-00822-t002:** Oxygen evolution reaction (OER) activity observed by mass activity analysis for the different metal-modified ZrP catalysts and benchmark catalyst IrO_x_ in alkaline electrolyte.

Metal	ZrP Support	Mass Activity at 1.58 V vs. RHE (A/g)	Reference
Co	α-ZrP	115	[87]
	ZrP rods	4	[87]
	ZrP cubes	72	[87]
	ZrP spheres	56	[87]
Ni	α-ZrP	50	[87]
	ZrP rods	85	[87]
	ZrP cubes	91	[87]
	ZrP spheres	272	[87]
IrO_x_	-	257	[91]

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
