# Peer review of "Preparation of Zirconium Phosphate Nanomaterials and Their Applications as Inorganic Supports for the Oxygen Evolution Reaction"

_nanomaterials, 2020, doi:10.3390/nano10050822_

Round 1

Reviewer 1 Report

This is a comprehensive review summarizing various preparation methods of nanostructured zirconium phosphate in the first part, and in the second part summarizing its potential application as a support for transition metals used to catalyze oxygen evolution reaction (OER).

When describing various method of ZrP nanoparticles preparation, it would be reader-friendly to include a Table with described method, obtained shape, size and reference, ideally arranged chronologically. 

The authors should also include XRPD patterns in their review, since they are reported in their recent paper under doi.org/10.1039/C9DT04135D

Fig 2., please include subscripts on H3PO4

Fig 10., there should be information in what media the LSV curves were taken and if the curved are iR compensated or not. The LSV curves are not fully visible because of a cut at 0 value. This should be corrected so the peak for Ni (II) to Ni(III) around 1.3-1.4 V vs. RHE, which is commonly observed can be seen. doi.org/10.1039/C5TA10317G

The catalytic activity towards OER was summarized by comparing overpotential at 10 mA/cm2.  It would be however more insightful if the authors include metal mass percentage activity and its comparison with the benchmark IrO2/C catalyst.

The overpotential for Co and Ni catalyst is pretty high. There are much better electrocatalysts reported. The authors should include comparison with Ni and Co of the same mass loading on different supports to understand if ZP is really beneficial for application as a support for OER.

The review does not address potential anodic oxidation of support.

The following reference should be included J. Chem. Sci. Vol. 127, No. 11, November 2015, pp. 1945–1955. c Indian Academy of Sciences.
DOI 10.1007/s12039-015-0955-2

Summarizing, this review is publishable after the above mentioned suggestions will be addressed.

Reviewer 2 Report

The authors reviewed how different ZrP nanomaterial work for OER having a title “Preparation of Zirconium Phosphate Nanomaterials and their Applications as Inorganic Supports for the Oxygen Evolution Reaction”. Before publication, following issues need to be addressed

  1. In the introduction part if possible please try to explain why the ZrP with different morphology can enhance OER activity and main advantage of focusing on it.
  2. You can explain general OER mechanism under alkaline or acidic conditions in the introduction part
  3. You can plot graphical abstract
  4. In the 2. Synthesis part makes sure after 2.2. sub point the others are mentioned correctly in the order(2.3. & 2.4.)
  5.  In the 3.1. please make a correction of 0.1 KOH electrolyte to 0.1 M KOH electrolyte
  6. If possible make a comparative table for OER activity depending on the different morphology by mentioning synthesis method for better analysis with proper references
  1. Other than the aforesaid correction, please revise the manuscript for any typo as well as grammatical errors

Round 2

Reviewer 1 Report

The authors improved the manuscript by addressing all the comments.

The manuscript can now be published.